

# The Inverted Microbial Loop Stimulates Mineralisation of Sedimentary Organic Detritus

Clare Woulds*[1], Dick van Oevelen[2], Silvia Hidalgo-Martinez[2, 3], Filip J. R. Meysman[2, 3]

[1]water@leeds, School of Geography, University of Leeds, Leeds, LS2 9JT, UK. ORCID 0000-0002-3681-1468

[2]NIOZ Royal Netherlands Institute for Sea Research, Department of Estuarine and Delta Systems, and Utrecht University,

Korringaweg 7, 4401NT Yerseke, the Netherlands. ORCID for DvO 0000-0002-1740-5317

[3] Excellence Centre for Microbial Systems Technology, Department of Biology, University of Antwerp, 2610 Wilrijk, Antwerp, Belgium. ORCID for FM 0000-0001-5334-7655, for SHM 0009-0005-1878-0361

*Correspondence to*: Clare Woulds (c.woulds@leeds.ac.uk)

**Abstract.** Respiration is a key process in the organic carbon cycle of marine sediments. The microbial community is considered the dominant actor in the overall sedimentary respiration, but knowledge is lacking about interactions with other components, particularly the macrofauna. The 'inverted microbial loop' hypothesis suggests that macrofaunal activity stimulates the microbial respiration of organic carbon through the mixing of fresh organic carbon to depth, and subsequent priming (i.e. activation of refractory detritus by co-respiration with fresh detritus). We conducted experimental incubations to partition

respiration amongst the microbial and macrofaunal components of the community and investigate interactions between them. We prepared sediment cores with native benthic communities, macrofauna only and microbial communities only. We added $^{13}C$ labelled fresh organic matter to these cores and measured respiration over 7 days, quantifying both $O_2$ consumption (reflecting remineralisation of all sedimentary organic C) and production of $^{13}C$ dissolved inorganic C (DIC, reflecting remineralisation of labile organic C).

Macrofaunal and microbial communities showed an approximately equal contribution to the total community respiration, while the fate of the added fresh organic C in different treatments suggested competition for this resource between macrofauna and microbes. Consumption of $O_2$, which reflected remineralisation of ambient as well as added fresh organic C, showed greater rates when macrofaunal and microbial communities were present together than the sum of their separate rates. This provides direct experimental evidence that the inverted microbial loop mechanism stimulates mineralisation of less reactive, ambient

organic C. The inverted microbial loop effect is likely to be enhanced following deposition of fresh organic C onto the seafloor, as occurs after a spring bloom.





## 1 Introduction

Marine sediments play a key role in the global carbon cycle, as they serve as the location for long-term burial of organic carbon (C), with shelf and deltaic settings being disproportionately important (Berner, 1982). After being deposited at the sediment

surface, organic C can follow two principal pathways: it may either be mineralized by the respiratory metabolism of resident organisms (macrofauna, meiofauna or microorganisms) or it may escape mineralization through burial into deeper sediment horizons and thus removed from the short-term carbon cycle involving atmosphere, terrestrial ecosystems and oceans. Isotope labelling experiments have shown that ~ 90% of the incoming organic matter is subject to respiration, and is released back to the water column as dissolved inorganic carbon (~80%) or dissolved organic carbon (~10%), while the remaining part is

eventually buried and preserved (Middelburg and Meysman, 2007; Burdige, 2007). Hence, an improved understanding of the factors that govern benthic metabolism and respiration is important to further our knowledge of marine carbon cycling.

Total community respiration – often measured as the flux of dissolved inorganic carbon (DIC) or oxygen ($O_2$) that crosses the sediment-water interface – is influenced by several external factors. Total community respiration increases with both temperature and organic matter deposition, and as a consequence, it tends to vary with season (Kristensen, 2000) and shows a

strong negative relationship with water depth (Middelburg et al., 2005; Stratmann et al. 2019). Strong current and wave activity can also induce high respiration rates in sandy sediments, as advective porewater exchange supplies both fresh organic matter and oxygen, thus stimulating mineralisation activity (Huettel et al., 2003; Erenhauss and Huettel, 2004; Alongi et al., 2011). Light availability has also been suggested to control respiration in shallow environments, as photosynthesis in biofilms at the sediment surface can increase the supply of organic C or oxygen to the sediment community (Kristensen, 2000; Middelburg

et al., 2005; Hubas et al., 2007).

In contrast to our knowledge of the external factors that govern total community respiration, we lack an understanding of the internal mechanisms that determine how respiration is partitioned amongst the different groups of organisms that make up the benthic community, and especially, how interactions between those groups can influence the total community respiration. The microbial component is often assumed to be of paramount importance in driving total community respiration, and evidence

for this comes from both observational (e.g., Schwinghamer et al., 1986; Hubas et al., 2006) and modelling (Van Oevelen et al. 2006) studies. Likewise, other studies have emphasized that macrofaunal activity may also play a major role (e.g. Herman





et al. 1999; Heip et al., 2001), either through their direct contribution to respiration, or through indirect interactions that stimulate the respiration by the microbial community.

For this reason, it is important to consider how interactions between macrofaunal and microbial activity may influence sediment respiration. In the water column, macrofauna and microbes have typically been linked through the 'microbial loop', in which organic C that is lost to the dissolved organic carbon (DOC) or dissolved inorganic carbon (DIC) pools is subsequently assimilated and transformed into new biomass by the microbial community, and becomes available once again to macrofauna as a food source (Kemp, 1988; 1990; Vasquez-Cardenas et al., 2020). This looping stimulates C cycling and increases remineralization efficiency. However, there is little evidence that macrofaunal grazing on microbes plays an important role in the microbial loop in sediments, as studies have shown that bacterial biomass is a rather minor food source for benthic faunal communities, which typically rely on an input of fresh algal detritus from the water column (e.g. Kemp, 1990; Van Oevelen et al., 2006; Guilini et al., 2009). As an alternative suggestion, Middelburg (2018) has proposed the 'inverted microbial loop' concept, which states that macrofaunal activity can actually stimulate sedimentary microbial activity and respiration, rather than depressing it by grazing. In this view, macrofauna transport freshly deposited organic matter to depth, thus making it available to the sediment dwelling microbes, which prompts a priming effect, i.e., an increase in the decomposition rate of native sedimentary organic carbon after fresh organic matter input.

From a conceptual point of view the 'inverted microbial loop' makes sense: it is well known that macrofauna can stimulate both the supply of the electron donor (fresh organic C) as well as the electron accepter ($O_2$) to the resident microbial community in marine sediments. Bioirrigation increases the oxygenated volume of sediment several-fold and supplies respiratory electron acceptors, stimulating microbial degradation (Aller and Aller 1998; Herman et al, 1999; Kristensen, 2000; Glud et al., 2003; Middelburg et al., 2005), enhancing total respiration by 25-271% (see Kristensen, 2000, and references therein). Likewise, solid particle mixing by macrofauna transports freshly deposited organic material to depth in the sediment, which brings together labile and refractory organic carbon. This enhanced supply of $O_2$ and/or fresh organic C could lead to priming, whereby refractory organic carbon is decomposed that would otherwise not have been mineralised. Still, the occurrence of priming seems very much dependent on the compounds and environment in question (Bengtsson et al., 2018), but it has previously been observed in marine sediments (van Nugteren et al., 2009; Gontikaki et al., 2015). Priming



mechanisms require further investigation, but are likely to involve changes to microbial population composition and activity, and associated enzyme production, mutalism and/or co-metabolism (Bianchi, 2011).

Stimulation of microbial processes by macrofaunal activity is also thought to have played a role in Earth evolution.

It has been proposed that the rise of animals around 540 Myr ago, and the concomitant evolution of burrowing and bioturbation, may have instigated a more efficient recycling of organic matter in the seafloor with potential Earth system impacts (Meysman et al. 2006). Recent studies have quantitatively explored this idea using Earth System Models, and propose that this effect may have been large enough to increase atmospheric $CO_2$ levels, inducing global warming and ocean anoxia (van de Velde et al, 2018).

Here we take an experimental approach to explicitly quantify the strength of the inverted microbial loop effect. To this end, we aim to partition the total respiration in marine sediments into contributions of microbial respiration, faunal respiration and a microbial-faunal interaction term in order to test and quantify the strength of the inverted microbial loop for both reactive and refractory organic C. Few studies have experimentally assessed the contribution of these three components to total sediment respiration. Previous attempts have taken a theoretical approach (e.g. Schwinghamer et al., 1986; Franco et

al., 2010), but these approaches do not account for positive interactions, such as the inverted microbial loop, between components of the benthic community. Furthermore, Van Nugteren et al. (2009a) found that the resource partitioning of fresh organic matter between macrofauna and microbes depends on the spatial distribution of the organic matter, with only microbes being able to efficiently utilise resources that are diffusely distributed throughout the sediment. This leads us to hypothesize that the inverted microbial loop effect may apply predominantly to the ambient, more refractory and 'diffusively' distributed

sedimentary organic C, and less to the fresh organic C that is concentrated on the sediment surface.

## 2 Methods

### 2.1 Experimental Approach and Rationale

Marine sediment cores were constructed and incubated over time. The total oxygen uptake (TOU) rate was measured as the indicator of total community respiration, primarily representing remineralisation of refractory organic C. In parallel, we

quantified fresh organic matter respiration (FOMR) in the same cores by addition of $^{13}$C labelled substrates and determining

the subsequent release of $^{13}$C labelled dissolved inorganic C (DIC).

To obtain insights into the TOU and FOMR of different components of the benthic community, and assess the interaction

between microorganisms and macrofauna, we applied the following four treatments when constructing experimental cores: 1)

Control: natural, intact sediment cores. Respiration is due to prokaryotes and macrofauna, and their interaction; 2) Defaunated:

sediment cores that were defaunated by inducing anoxia, and exposed again to overlying oxygenated water. Respiration is

dominated by prokaryotes (with some meiofauna present), but macrofauna are excluded 3) Restocked: sediment cores were

first de-faunated (by inducing anoxia), and then exposed again to overlying oxygenated water and re-stocked with a controlled

macrofaunal community. Respiration is due to prokaryotes and a controlled biomass of macrofauna, and their interaction. 4)

Fauna: sediment cores were constructed that contain only clean construction sand, to which macrofauna were introduced.

Respiration is due to macrofauna. A control with only clean construction sand was run, but TOU data was not acquired due to

instrument problems. However, we expect microbial respiration to be small in these construction sand cores compared to that

of the macrofauna added.

The experiment with the four treatments was conducted twice using different $^{13}$C labelled substrates. In a first

experiment, $^{13}$C labelled algal detritus from an axenic culture (13C-AA) was added, which allowed tracing of C into the

microbial biomass. In the second experiment, we added natural microphytobenthos cultured in the presence of $^{13}$C labelled

bicarbonate(13C-MPB), thus providing a fully natural fresh C source.

If there were no interactions between components of the benthic community, respiration rates measured in the 'fauna'

treatment can simply be added to those from the 'defaunated' treatment and would equal the rates measured in the re-stocked

treatment (macrofauna and microbes + meiofauna together). Deviations from this expectation are indicative of positive (i.e.

inverted microbial loop) or negative (i.e. competitive) interactions between components of the benthic community.

## 2.2 Sediment Collection and Experimental Conditions

Experiments were conducted in June 2010 and June 2011 at the Netherlands Institute for Sea Research (Yerseke, The

Netherlands). Sediment cores and filtered seawater were collected from nearby intertidal sites in the Oosterschelde estuary.

Key experimental details are listed in Table 1. Surface sediment cores (19.4 cm inner diameter) were collected in acrylic tubes,




and after a short transit to the laboratory (< 2 hr), they were kept in darkness in a climate-controlled room at ambient

temperature with overlying filtered seawater (0.2 µm pore size) at *in-situ* salinity (Table 1). Overlying water was oxygenated

using air stones, except for when periods oxygen consumption rates were determined (see description below).

| Experiment | Axenic Algae (13C-AA) | Natural Microphytobenthos (13C-MPB) |
|---|---|---|
| Site Latitude/Longitude | 51.553963°N, 3.874659°E | 51.471944°N, 4.063889°E |
| Date (sample collection and incubation experiment) | June 2010 | June 2011 |
| Core inner diameter [cm] | 19.4 | 14.3 |
| Temperature [°C] | 19 | 17 |
| Added C dose [mg C m$^{-2}$] | 395 ± 11 | 1730 ± 204 |

**Table 1. Details of sampling sites and experimental conditions.**

### 2.3 Experimental Treatments

Three replicate cores were subjected to each of the 4 treatments in the two experiments. De-faunation of sediment cores for

the defaunated and restocked treatments was conducted by asphyxiation as in described Rao et al. (2014), which leaves the

sediment stratification intact (as opposed to defaunation by sieving). To this end, anoxic conditions were induced by purging

the overlying seawater in the core with N$_2$ gas for several hours and then sealing the cores with gas-tight lids for 4-6 days.

After this anoxic period, the cores were opened and the overlying water was exchanged and re-aerated with air stones. Dead

organisms that had migrated to the sediment surface were first removed with tweezers. The cores were subsequently left

undisturbed for one day to allow the re-oxidation of reduced compounds that had accumulated in the surface layer of sediment.

After one day of reaeration, a mix of fauna (Table 2 and further below) was added at the surface of restocked treatment cores,

and were allowed migrate into the sediment. Cores were then acclimated again for 1-2 days before being amended with $^{13}$C

labelled organic detritus. After that the cores were incubated for 7 days.

| Species | Axenic Algae (13C-AA) (g wet weight m$^{-2}$) | Microphytobenthos (13C-MPB) (g wet weight m$^{-2}$) |
|---|---|---|
| *Arenicola marina* | 263.9 ± 27.1 | 274.0 ± 37.4 |
| *Hediste diversicolor* | 44.0 ± 3.4 | 62.3 ± 24.9 |
| *Cerastoderma edule* | 358.6 ± 54.1 | 386.1 ± 49.8 |



| | | |
|---|---|---|
| *Heteromastus filiformis* | 10.2 ± 3.0 | 12.5 ± 5.6 |
| *Hydrobia ulvae* | 152.3 ± 3.4 | 143.2 ± 5.0 |
| Total biomass | 828.9 ± 91.4 | 878.0 ± 124.5 |

**Table 2. Biomass of macrofaunal taxa added (g wet weight m⁻²) in the 'restocked' and 'fauna only' treatments. Note that the same biomass values was aimed for in the two treatments, so means and standard deviations are reported across both treatments.**

Based on background knowledge about the sampling site (Daggers et al., in press), we knew *a priori* that the fauna at the sampling location predominantly consists of the polychaetes *Hediste diversicolor*, *Arenicola marina* and *Heteromastus filiformis*, the gastropod *Hydrobia ulvae* and the bivalve *Cerastoderma edule*. These species were therefore selected for the re-

stocked and fauna treatment and introduced into cores at densities that simulated the natural faunal community (Table 2 and results).

### 2.4 Experimental Procedures

For the "axenic algae" (13C-AA) experiment, the marine diatom *Skeletonema costatum* was axenically cultivated in $^{13}$C-labelled medium. The resulting algal cells were 28.25 and 14.49 atom % $^{13}$C for two separate batches. A slurry of freeze-dried,

$^{13}$C-labelled biomass (395 ± 11 mg C m⁻²) was carefully mixed into to the water column and allowed to settle onto the sediment-water interface (so that the whole surface area was more or less homogeneously covered with labelled substrate).

For the "microphytobenthos" (13C-MPB) experiment, microphytobenthos was collected at the study site at the same time as the sediment cores. The top millimetres of sediment were scraped off at locations where distinctly brown patches (indicative of high MPB biomass) were present. This sediment/microphytobenthos mixture was enriched with $^{13}$C through incubation in

a white plastic culture box (0.6 m x 0.4 m) that was placed outside (ambient temperature) and covered with a transparent lid (natural light). The thin layer of sediment in the culture box was topped with a thin layer of ambient seawater (~5 mm) to prevent dehydration. The next day, 0.136 g of $^{13}$C-labelled sodium bicarbonate (NaH$^{13}$CO$_3$, 99%; Cambridge Isotope Laboratories) was dissolved in 50 ml of filtered seawater and introduced into the culture. This label addition was repeated daily for 7 days, after which the labelled microphytobenthos was harvested by scraping off the top several millimetres. This

mixture was homogenised, frozen in liquid nitrogen (to kill the MPB cells and prevent respiration activity by MPB during the sediment core incubations) and stored until further usage at -18ºC. The chlorophyll-a concentration of this slurry was

determined on 3 subsamples using standard fluorometry methods (Aminot and Rey, 2001). The resulting concentration ($37 \pm 5$ ug g$^{-1}$) was converted to C using a conversion factor of 40 (Stephens et al., 1997) resulting in an estimated $1.5 \pm 0.2$ mg C g$^{-1}$. Cores were amended with 12.5 cm$^3$ of slurry (density 2.0 g cm$^{-3}$), which was added using a pipette, and allowed to settle

onto the sediment surface over several hours. Each core hence received 3.08 mmol of C from MPB (corresponding to 2.30 g C m$^{-2}$). The $^{13}$C labelling level of the MPB was unknown, but this does not prevent calculation of respiration rates from the measured $^{13}$C-DIC production.

In both experiments, cores were incubated for 7 days after addition of labelled algae, with repeated measurements of O$_2$ consumption and $^{13}$C-DIC release during this period (see below). At the termination of the experiment, sediment cores were

sub-sampled using plastic syringes and samples were frozen at -18°C. The remaining sediment in each core was sieved through a 1 mm mesh. Fauna retained on the mesh were picked, and their wet biomass was recorded, after which specimens were frozen for further analysis.

### 2.5 Respiration Measurements

Benthic respiration was measured in all cores through total O$_2$ uptake (TOU, i.e. proxy for total community respiration,

primarily of refractory organic C) and release of $^{13}$C=DIC (i.e. proxy for respiration of fresh, labile algae) at several time points: before and straight after addition of isotopically labelled algae, and every 1.5 days for 7 days thereafter.

At the beginning of each respiration measurement, the overlying water of each core was sampled for dissolved oxygen (DO), dissolved inorganic carbon (DIC), and $^{13}$C of DIC. Cores were then sealed with custom-built gas-tight lids, excluding all air bubbles, and incubated for 2-5 h until O$_2$ saturation in the overlying water had fallen to ~70%. During the closed incubation,

core top water was stirred continuously. At the end of each respiration measurement, core top water was again sampled for the parameters listed above. After respiration measurements, the overlying water in each core was exchanged to avoid build-up of (toxic) metabolic products, and kept aerated by gentle bubbling with air.

### 2.6 Analytical

Samples for O$_2$ analysis were collected in glass Winkler bottles with ground glass stoppers and known volumes. Bottles were

allowed to overflow copiously before MnSO$_4$ and KI in KOH solutions were added and stoppers inserted. Samples were shaken

for 30 s, and stored in at 4ºC before analysis within 2 days. Samples were titrated against standardised thiosulphate solution using a micro-titration set-up.

Dissolved inorganic carbon (DIC) samples (20 ml) were stored in crimp-cap vials, and preserved with $HgCl_2$ (20 µl of saturated solution). Vials were stored at 4°C, inverted with the caps standing in water to prevent the exchange of $CO_2$ with the

atmosphere. Samples were analysed for DIC concentration and $\delta^{13}C$ as detailed in Moodley et al. (2000), using a Carlo Erba MEGA 540 gas chromatograph coupled to a Finnigan Delta S isotope ratio mass spectrometer, following creation of a He headspace in each sample vial. Standards used were acetanilide, and the IAEA standard CH-6. Repeat analysis of standard materials yielded precision of ± 4.4% for DIC concentrations, and ± 0.09 ‰ for $\delta^{13}C$.

Sediment samples from the axenic algae experiment were analysed for $^{13}C$ incorporation into bacterial phospholipid fatty acids

(PLFAs) using a modified Bligh-Dyer extraction after Middelburg et al. (2000). Lipids were extracted at room temperature in a mixture of chloroform, methanol and water, before being loaded onto silicic acid columns. Phospholipid fatty acids were eluted in methanol and derivatised to fatty acid methyl esters (FAMEs) using methanolic NaOH. The C12:0 and C19:0 FAMEs were used as internal standards. Samples were separated by gas chromatography  using a BPX70 column, combusted in a Thermo GC combustion II interface, and isotopic ratios were measured using a Thermo Delta + isotope ratio mass

spectrometer.

### 2.7 Data Analysis

The Total Oxygen Uptake (TOU) of the sediment was calculated from the difference in the total amount of dissolved $O_2$ present (i.e. $O_2$ concentration x chamber volume) between the start and end of each closed incubation, divided by the time elapsed in each measurement ($\Delta t$), and normalised to the surface area of the cores (SA), i.e., TOU = ($O_2$end – $O_2$start)/$\Delta t$ /SA.

Release of $^{13}C$-DIC was determined from the difference in total amount of $^{13}C$ in each chamber (i.e. DIC concentration x chamber volume x At% $^{13}C$ DIC) between the start and end of the incubation, divided by the duration of the incubation ($\Delta t$), and normalised to the surface area (SA) of the cores, i.e. $^{13}C$-DIC Release = ($^{13}C$ end – $^{13}C$ start)/$\Delta t$ /SA.

Cumulative TOU and $^{13}C$-DIC release were calculated by multiplying each of the measured rates described above by the time periods between closed TOU/$^{13}C$ incubations. These were then summed to produce estimates of cumulative TOU and $^{13}C$-DIC

release over the whole experiment for each treatment.



Uptake of $^{13}$C into bacterial biomass in the 13C-AA experiment was calculated by first subtracting naturally present $^{13}$C based on analysis of unlabelled sediment. Presence of $^{13}$C in the bacterial indicators i-C14:0, i-C15:0, ai-C15:0 and i-C16:0 was then summed, and scaled up based on these compounds representing 14% of total bacterial PLFAs, and PLFAs representing 5.6% of total bacterial biomass (Boschker and Middelburg, 2002).

Statistical analysis of data was performed using Minitab 18.

## 3 Results

### 3.1 Biomass of Fauna

The living macrofaunal biomass recovered from the control treatment at the end of the 13C-AA experiment (4.8±3.2 g wet weight per core) was far greater than that recovered from the defaunated treatment (0.9±1.0 g wet weight per core). This

illustrates that asphyxiation removed >80 % of the fauna, but still a restricted anoxia-tolerant community (Hediste, Arenicola) survived. The natural biomass present (in the control treatment) was lower than anticipated, and so biomass added to the restocked and fauna treatments (18-21 g wet weight per core, Table 3) was four times higher than the control treatment. Very few dead organisms were seen in treatments where fauna were added, with the majority recovered alive at the end of the experiment. Note that macrofauna biomass data were not recorded for the 13C-MPB experiment.

| Experiment | Treatment | Recovered Biomass (g wet weight per core) | Percentage Recovery |
|---|---|---|---|
| 13C-AA | Control | 4.8 ± 3.2 | N/A |
| | Defaunated | 0.9 ± 1.0 | N/A |
| | Restocked | 21.1 ± 1.4 | 86 ± 6 % |
| | Fauna | 17.8 ± 5.4 | 73 ± 22 % |

**Table 3. Biomass of macrofauna recovered from cores at the end of the 13C-AA experiment. Values are means ± standard deviation for n = 3 replicates. Data not available for the 13C-MPB experiment.**

### 3.2 Total Oxygen Uptake

Total Oxygen Uptake rates showed substantial variation and ranged from 9-91 mmol $O_2$ m$^{-2}$ d$^{-1}$ in the 13C-AA experiment, and 7-241 mmol $O_2$ m$^{-2}$ d$^{-1}$ in the 13C-MPB experiment (Fig. 1). TOU values were generally higher in the 13C-MPB





experiment compared to the 13C-AA experiment. Due to problems with the oxygen measurement technique, data is lacking

for the control and fauna treatments in the 13C-AA experiment during the first 2 days after feeding.

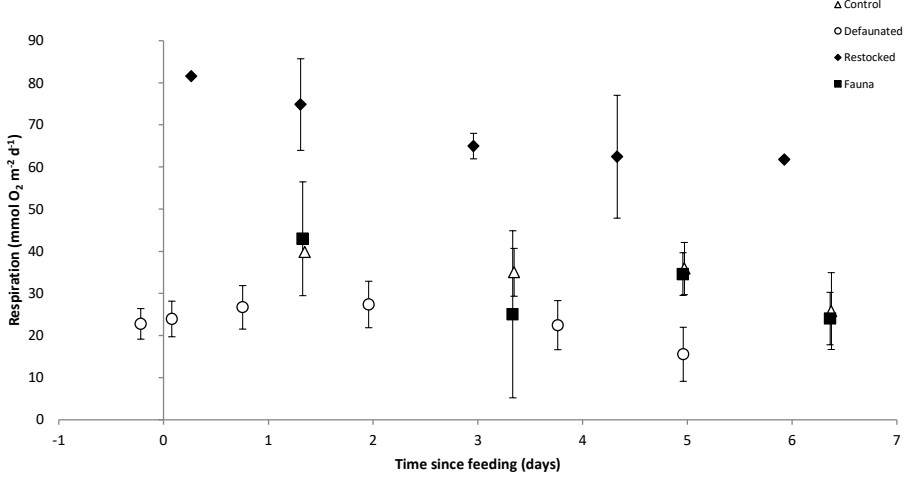

(a)





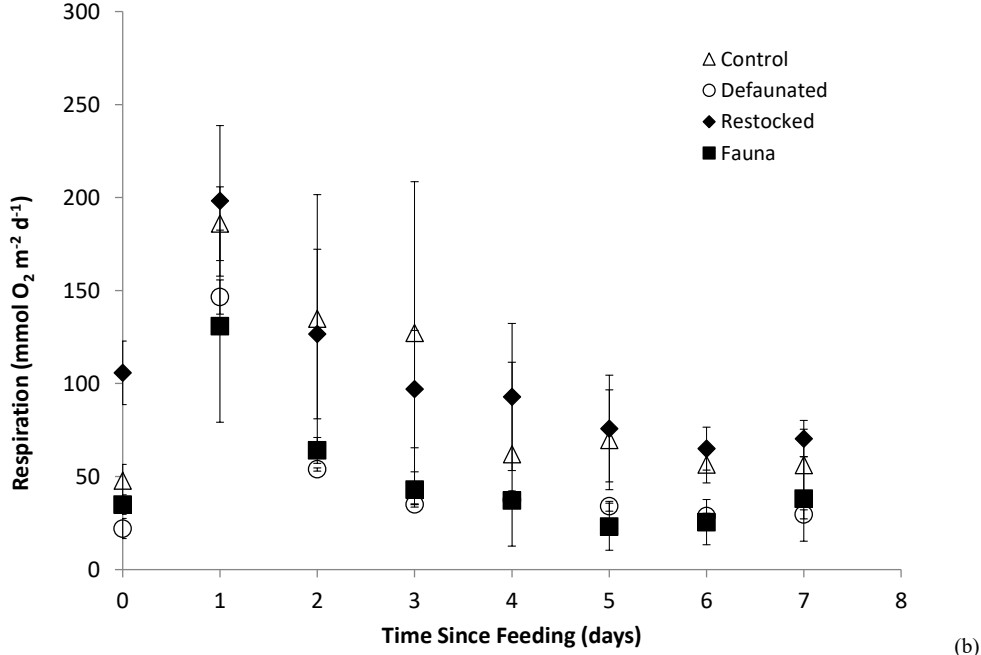

(b)

**Figure 1. Respiration rates, determined as the Total Oxygen Uptake (TOU) of the sediment, in A) the 13C-AA experiment and B)**
**the 13C-MPB experiment.**

In the 13C-AA experiment, TOU showed a slight decrease over time in the restocked treatment, but no clear temporal pattern

in the other treatments (Fig. 1A). In the 13C-MPB experiment all treatments displayed a similar temporal pattern, with maximal

TOU values immediately after algal addition, and TOU values returning to pre-feeding levels after ~6 days (Fig. 1B).

Differences in TOU between treatments were apparent in both experiments (Kruskal-Wallis $p < 0.001$ for both experiments).

In the 13C-AA experiment, TOU values were always higher in the re-stocked cores compared to other treatments (Mann-

Whitney pairwise comparisons $p < 0.001$). There was also a significant difference between the control and defaunated

treatments, while other pairs of treatments were not significantly different (Mann-Whitney pairwise comparisons $p = 0.004$,

0.62, and 0.012 for control vs. defaunated, control vs. fauna, and defaunated vs. fauna, respectively). In the 13C-MPB

experiment rates were higher in the control and re-stocked treatments than in the defaunated and fauna only treatments





(Kruskal-Wallis, p≤0.001). TOU values in the control and restocked treatments (Mann-Whitney, p=0.130) and in the

defaunated and fauna only treatments (Mann-Whitney, p=0.516) were not significantly different from each other.

The cumulative TOU (i.e. the total $O_2$ consumed during each 7-day experiment) was higher in the 13C-MPB experiment

compared to the 13C-AA experiment. Cumulative TOU showed a similar pattern between treatments in both experiments (Fig.

2A) and was maximal in the restocked treatment, then followed by the control, and finally the defaunated and fauna only

treatments (Fig. 2A). Due to the high variability, significant differences between treatments could be identified for the 13C-

AA experiment (ANOVA, p < 0.001, groupings shown in Fig. 2A), but not for the 13C-MPB experiment (ANOVA, p = 0.052).

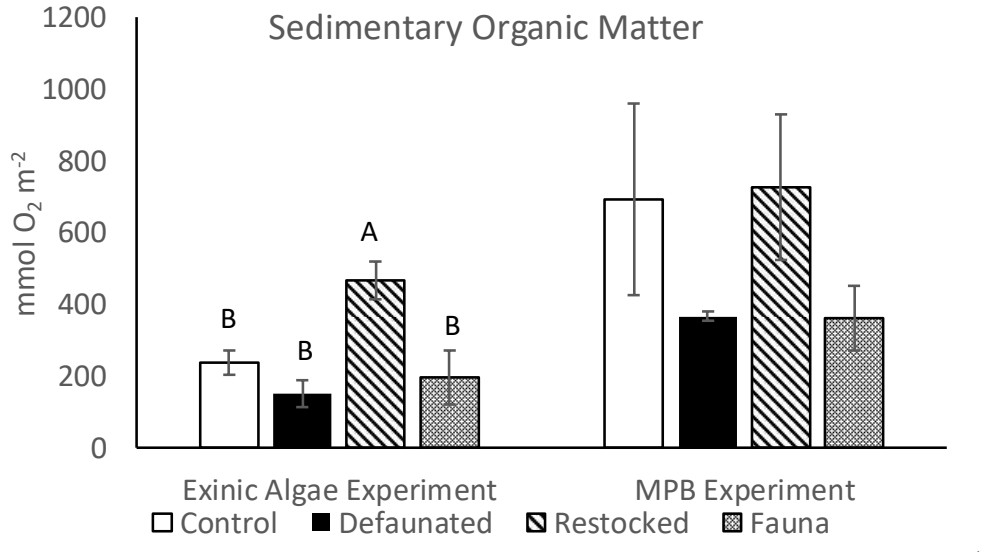

(a)



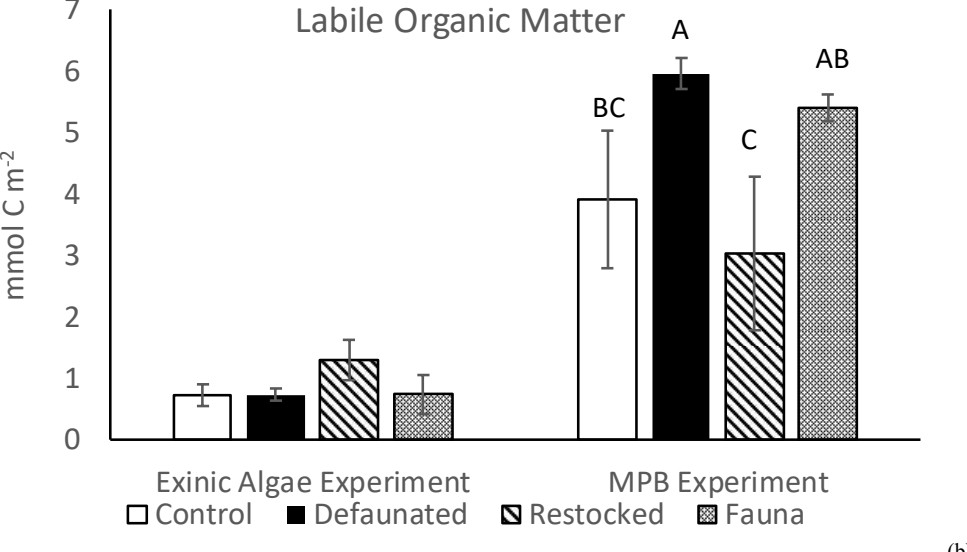

(b)

**Figure 2. Cumulative respiration over the whole of each experiment of A) total sedimentary organic matter, measured as Total Oxygen Uptake, and B) of added, fresh organic matter, measured as $^{13}$C-DIC release. Error bars are ± 1 standard deviation. Letters indicate significantly different treatments as shown by ANOVA.**

### 3.3 $^{13}$C-DIC Release

Fresh organic matter respiration (FOMR) rates were measured as the release of $^{13}$C -DIC and ranged between 0.04 – 1.85 mmol C m$^{-2}$d$^{-1}$ in the 13C-AA experiment, and 0.01 – 4.38 mmol C m$^{-2}$d$^{-1}$ in the 13C-MPB experiment (Fig. 3). Fresh organic matter respiration rates were substantially higher in the 13C-MPB experiment, but generally showed a similar time evolution in both experiments. Rates were always highest immediately after feeding, and declined rapidly thereafter, reaching constant levels after ~5 days (Fig. 3). Differences between treatments were most apparent during the first 2 days after feeding. For the 13C-AA experiment, the re-stocked and fauna treatments showed slightly higher initial FOMR rates (Fig. 3A). For the 13C-MPB experiment, the defaunated treatment showed higher initial rates (Fig. 3B). Due to the marked change in rates over time, significant differences in rates between treatments were only apparent on individual days. Significant differences between treatments were present 5 days after feeding in the 13C-AA experiment (Kruskal-Wallis p=0.04), and 1, 2 and 7 days after



feeding in the 13C-MPB experiment (Kruskall-Wallis, p=0.029, 0.038 and 0.034, respectively). However, pairwise Mann-Whitney U tests were not sufficiently powerful to show which pairs of treatments were significantly different on those days.

The cumulative FOMR was higher in the 13C-MPB experiment by a factor ~2-8 compared to the 13C-AA experiment for

different treatments, and showed different patterns in the two experiments (Fig. 2B). In the 13C-MPB experiment cumulative FOMR was maximal in the defaunated and fauna only treatments (ANOVA, p = 0.011, groupings shown in Fig. 2B). In the 13C-AA experiment there was no significant difference in cumulative FOMR between treatments (ANOVA, p = 0.061).

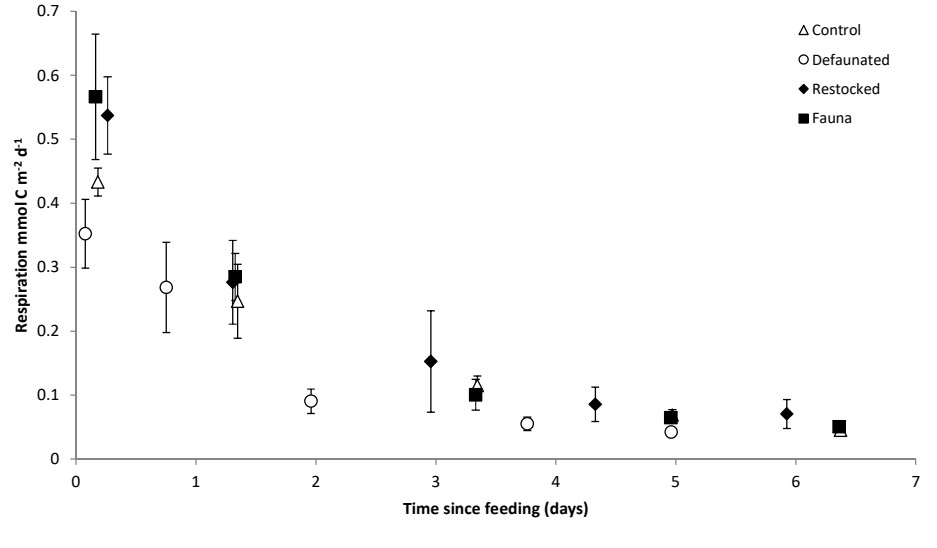

(a)



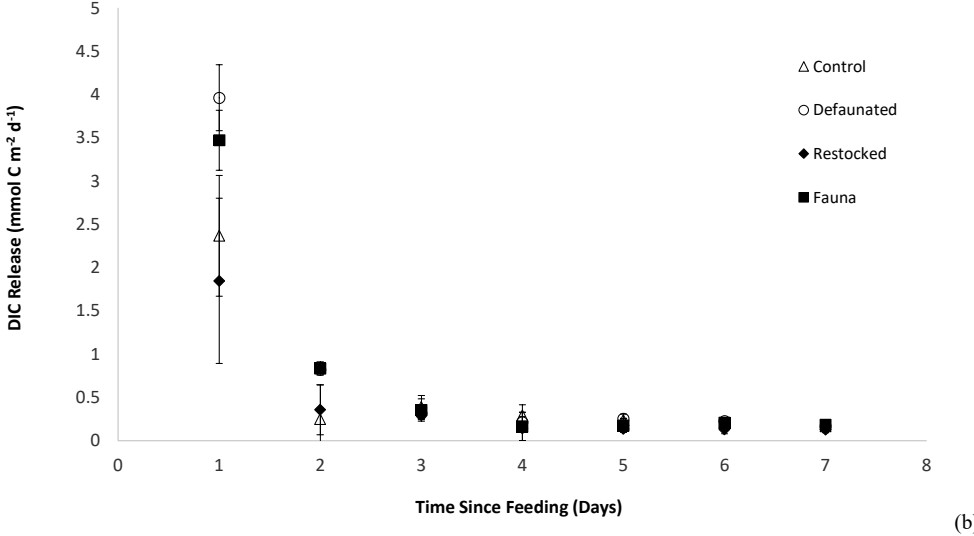

(b)

**Figure 3. Fresh Organic Matter Respiration calculated from $^{13}$C-DIC release for A) the 13C-AA experiment, and B) the 13C-MPB experiment.**

**3.4 Ratio of Oxygen Consumption versus $^{13}$C-DIC Production**

For each time point the TOU/FOMR ratio was calculated. The ratio ranged from ~50-1100 for the 13C-AA experiment and from ~19-958 for the 13C-MPB experiment (Fig. 4). There was a significant difference in TOU:FOMR ratios between the treatments in the two experiments (13C-AA experiment ANOVA, p=0.014; 13C-MPB experiment Kruskall-Wallis, p<0.001). Post-hoc testing showed that for the 13C-AA experiment the restocked treatment had significantly higher ratios than the defaunated treatment, and that the other two treatments were not significantly different from any other. Further, the fauna treatment, although not being statistically significantly different, appeared most similar to the defaunated treatment (Fig. 4). Similarly, in the 13C-MPB experiment, all treatments were significantly different from each other (Mann-Whitney, p <0.001-0.002), except for the defaunated and fauna treatments, which were not significantly different (Mann-Whitney, p=0.948).



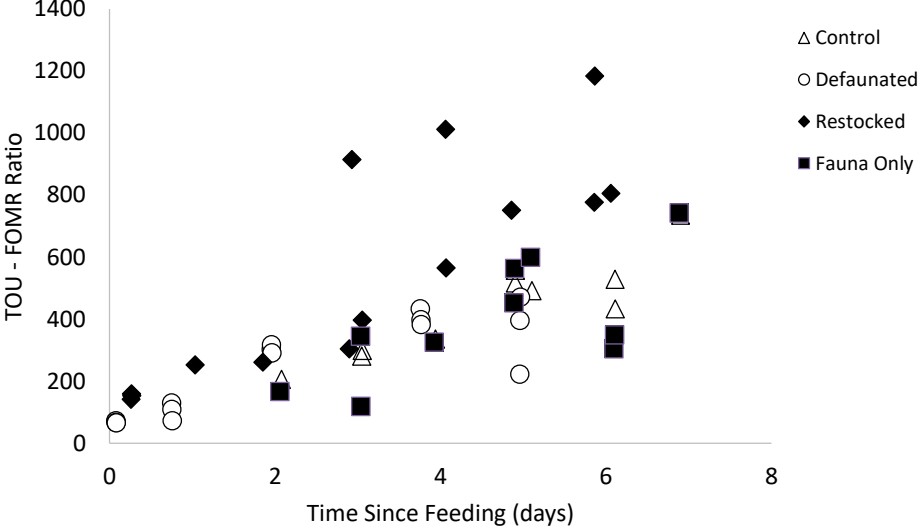

(a)

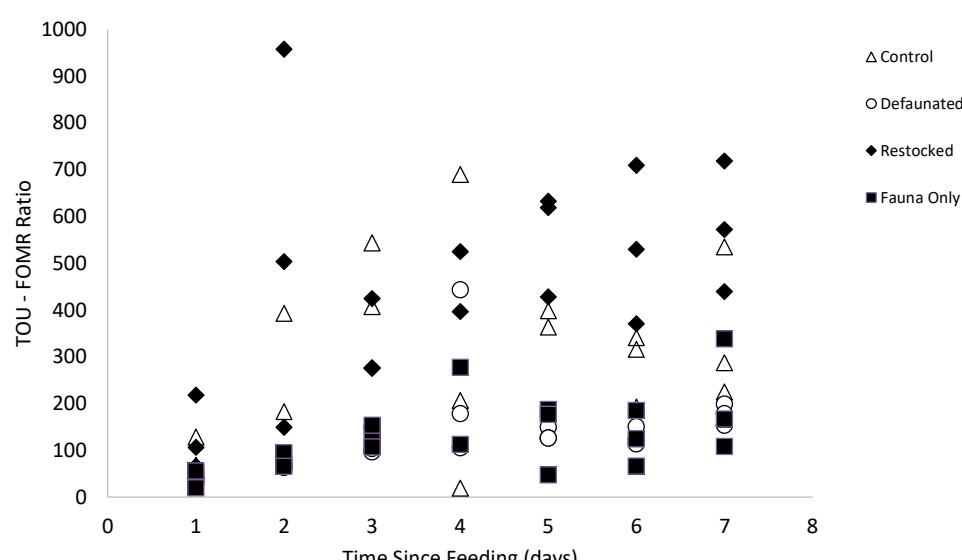

(b)





**Figure 4. The ratio between sediment O$_2$ uptake (TOU) and $^{13}$C-DIC release (FOMR) over time in A) the 13C-AA experiment and B) the 13C-MPB experiment. Measurements from individual replicates are plotted as separate points.**

**3.5 Bacterial Carbon Uptake**

Uptake of $^{13}$C into bacterial biomass was quantified by PLFA analysis in the 13C-AA experiment (Fig. 5) and predominantly occurred in the surface sediment (0-1 Cm), with uptake values 10-fold higher than the subsurface sediment (9-10 cm). Differences were notable between treatments: $^{13}$C uptake into bacterial biomass was not detectable in the fauna-only treatment, and ranged up to a maximum of 0.052 mg C g$^{-1}$ of wet sediment in the defaunated treatment (Fig. 5). Bacterial $^{13}$C uptake appeared to be maximal in the defaunated treatment (Fig. 5), but due to high variability in the control treatment, the observed differences between the control, defaunated and restocked treatments were not statistically significant.

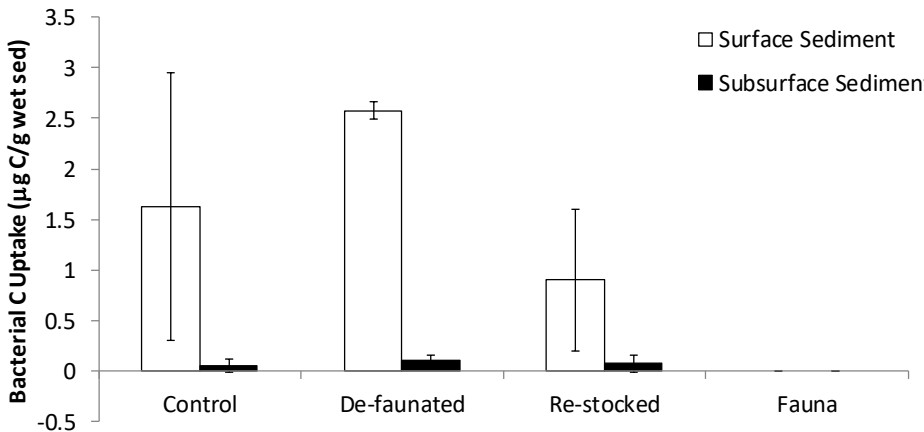

**Figure 5. Bacterial $^{13}$C uptake in surface and deep sediments in the axenic algae experiment. $^{13}$C labelled PLFAs were not detectable in samples from the fauna treatment.**





**3.6 Faunal Carbon Uptake**

Uptake of $^{13}$C into macrofaunal biomass was quantified in both the 13C-AA and 13C-MPB experiments and varied between
the two experiments (Fig. 6). All taxa showed uptake of labelled fresh organic matter in both experiments, providing $\delta^{13}$C
values up to 786 ‰ in the 13C-AA experiment and up to 286 ‰ in the 13C-MPB experiment (Fig. 6).

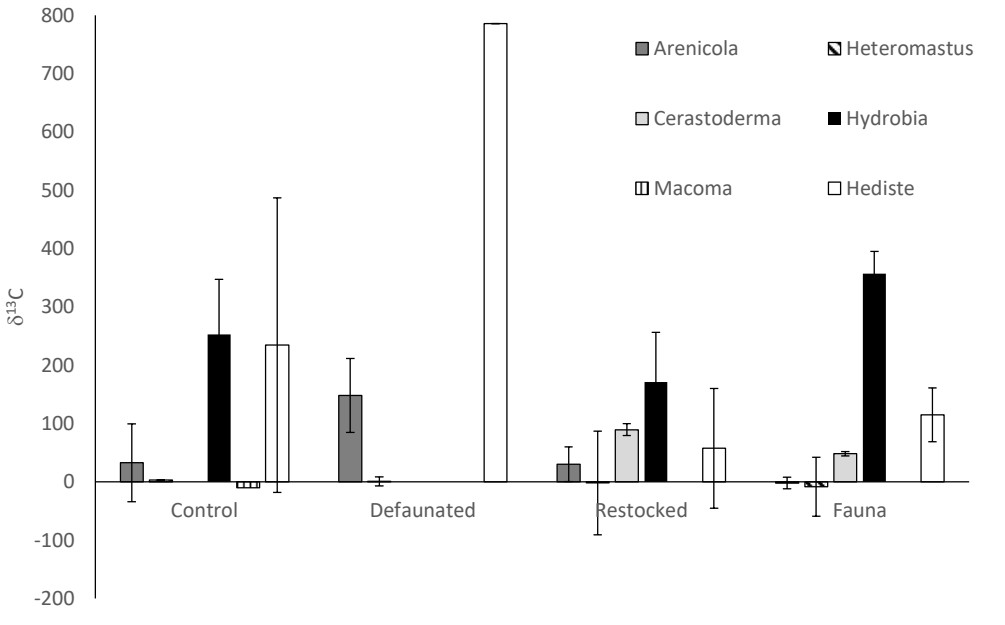

(a)



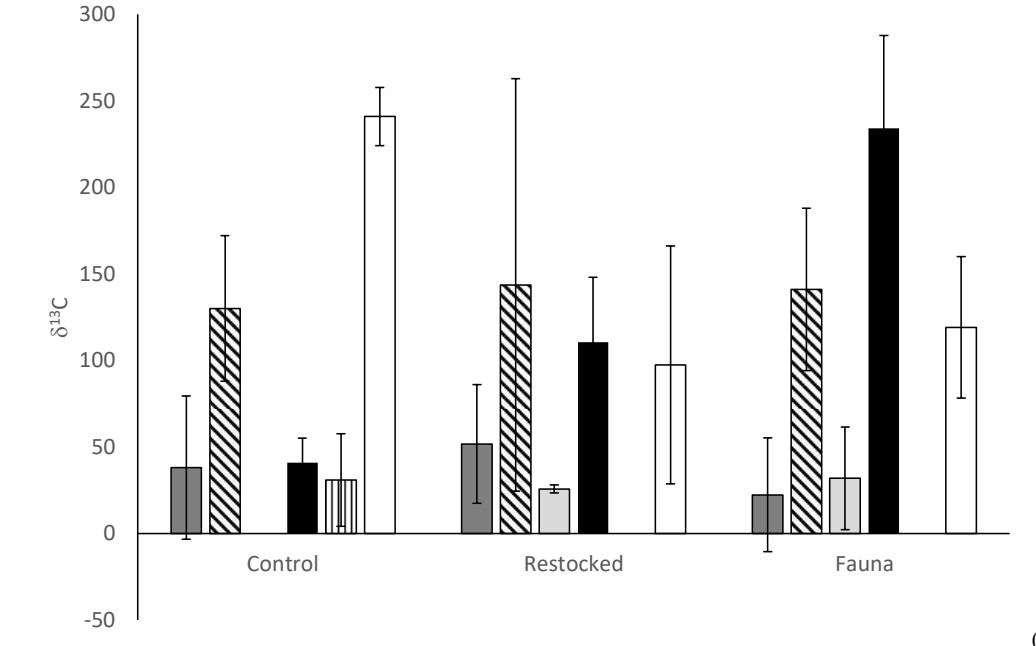


(b)

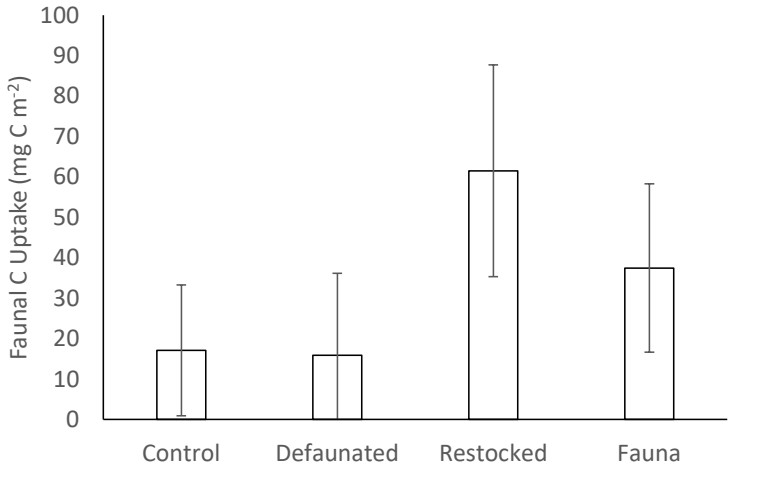

(c)




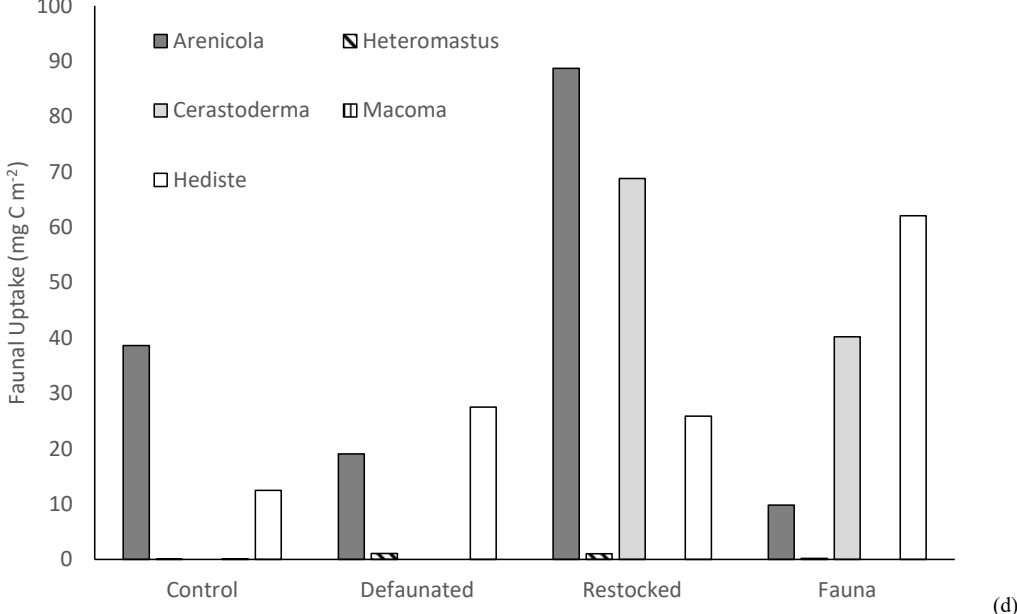

(d)

**Figure 6. Mean $^{13}$C isotopic signatures of macrofaunal taxa recovered from each treatment in A) the 13C-AA experiment, and B) the 13C-MPB experiment. The magnitude of faunal $^{13}$C uptake in the 13C-AA experiment as C) total faunal uptake, and D) by taxon.**
**Data for Hydrobia were excluded from panels C and D due to uncertainties regarding biomass. Bars represent mean ± 1 standard deviation.**

It should be noted that the C dose used in the two experiments varied (395±11 and 2300 mg C m$^{-2}$ for the 13C-AA and 13C-MPB experiments, respectively), and therefore direct comparison of faunal C uptake or labelling intensity between experiments is not possible. However, comparisons can be made regarding relative labelling levels of different taxa within each experiment,

and these showed significant differences in labelling between taxa (Kruskal-Wallis, p < 0.001 for the 13C-AA experiment and ANOVA, p < 0.001 for the 13C-MPB experiment). *Hydrobia ulvae* and *Hediste diversicolor* showed the highest labelling in both experiments, consistent with their high motility and surface deposit feeding habits. In contrast, the sessile and deep-living taxa *Arenicola marina, Cerastoderma edule* and *Macoma balthica* showed a lower labelling intensity (Fig. 6). Data for *Heteromastus filiformis* illustrated how variable the feeding can be within a single macrofaunal taxon, with low labelling in

the 13C-AA experiment, and high labelling in the 13C-MPB experiment. This may be due to a feeding preference by

*Heteromastus filiformis*, or could be a result of differences between the experiments in terms of C dose or other site-specific factors.

For the 13C-AA experiment, the wet weight of the macrofauna were measured, allowing quantification of total added C uptake by the macrofauna. *Hydrobia ulvae* was excluded from this calculation due to uncertainties in wet weight data. Macrofaunal

C uptake ranged from 15.9 mg C m$^{-2}$ in the defaunated treatment up to 61.5 mg C m$^{-2}$ in the restocked treatment (Fig. 6C). Macrofaunal uptake was generally higher in the restocked treatment than in other treatments, however variability in faunal biomass meant the differences were not statistically significant (Kruskal-Wallis, p = 0.192). Further, when total C uptake data from the 13C-AA experiment were pooled by taxon, there was a significant difference in uptake accounted for by different taxa (Kruskal-Wallis, p=0.044), with *Arenicola marina* and *Hediste diversicolor* each showing significantly more C uptake

than *Heteromastus filiformis*, and *Macoma balthica* (Mann-Whitney, P = 0.027 – 0.030, Fig. 6D).

## 4 Discussion

### 4.1 Total Oxygen Uptake and The Inverted Microbial Loop

The 'inverted microbial loop' hypothesis, originally proposed by Middelburg (2018), suggests that macrofaunal activity stimulates the microbial community by mixing freshly deposited, bioavailable organic carbon in deeper sediment horizons, thus increasing its availability to microbes for their metabolism. Therefore, the master response variable in the inverted

microbial loop concept is the respiration of organic matter, measured in our experiments as TOU. Total oxygen uptake reflects the respiration of the total sedimentary organic carbon pool, which in our experiments included both the slow-decaying ambient organic matter, as well as the fast-decaying fresh organic detritus that was added and carried the $^{13}$C label.

Interactions between components of the benthic community are indicated by differences between the TOU rates measured in

the restocked treatment (macrofauna plus microbes), and the sum of those in the defaunated (microbes only) and the fauna (macrofauna only) treatments. Our results indicate a positive interaction, as the sum of TOU in the defaunated and fauna only treatments tended to be less than the TOU of the restocked treatment. This was the case for all days except day 1 in the 13C-MPB experiment, and for the cumulative TOU in the 13C-AA experiment (Fig. 7). In these cases, the co-presence of macrofauna alongside the microbial community enhanced the TOU, thus suggesting an effect of the inverted microbial loop.



Furthermore, the cumulative TOU (cTOU) during each experiment was maximal in the control and restocked treatments, where

macrofaunal and microbial communities were present together (Fig. 2). In the 13C-AA experiment the cTOU in the restocked

treatment was approximately 2-fold higher than that in the fauna and defaunated treatments, and for the 13C-MPB experiment

both the restocked and control treatments were ~2-fold higher than the fauna or defaunated treatments (Fig. 2). In summary,

for the majority of our timepoints, the observed $O_2$ consumption rates, which reflect degradation of total sedimentary organic

matter, supported the occurrence of the inverted microbial loop.

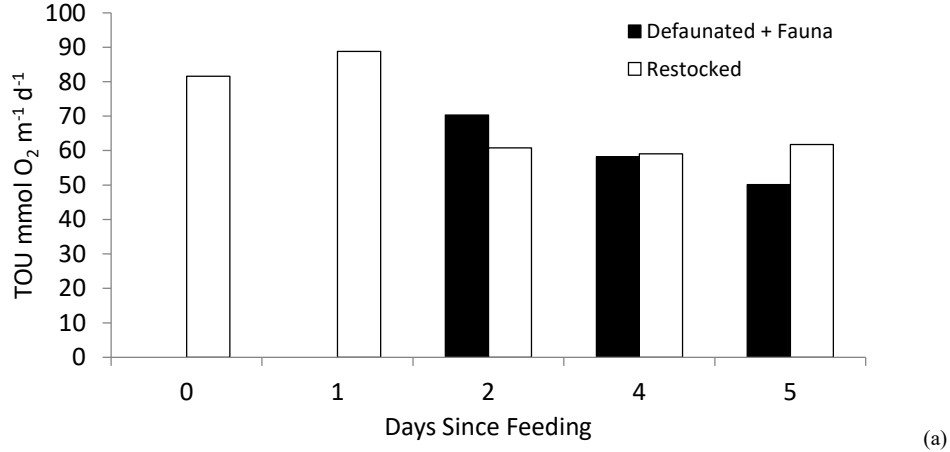

(a)



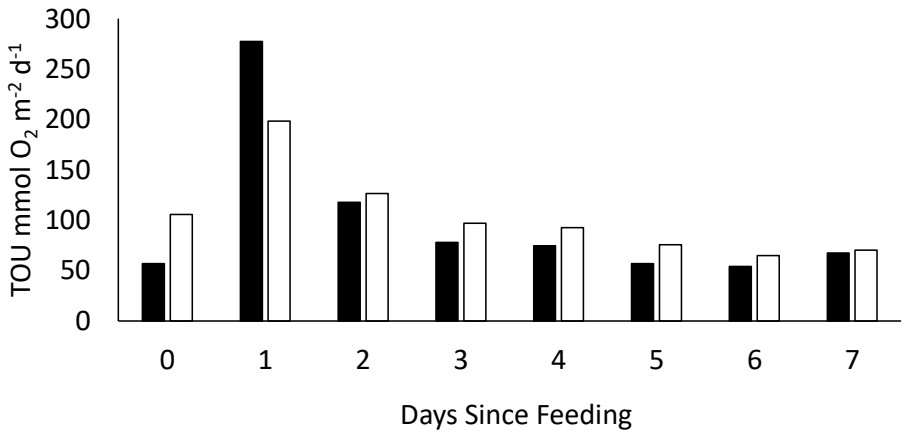

(b)

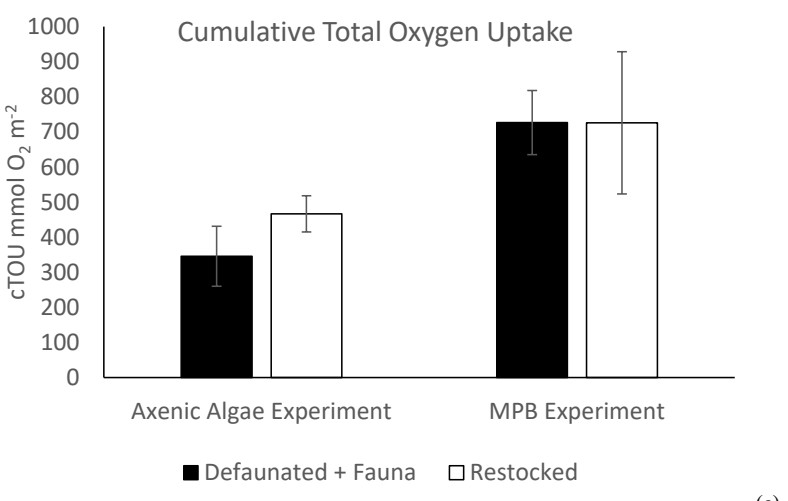

(c)

**Figure 7. Comparison between the sum of de-faunated and fauna rates and re-stocked TOU rates for the A) 13C-AA and B) 13C-**
**MPB experiments, and C for cumulative TOU (cTOU).**

The mechanisms behind the inverted microbial loop are relatively well documented. Macrofauna stimulate microbial activity

by enhancing the supply of $O_2$ via bioirrigation (Aller and Aller, 1998), as well as through the niche structuring and resource

partitioning that result from redistribution of organic matter to deeper sediment layers resulting from particle biomixing, thus



increasing the availability of organic matter to microbes (Schwinghamer et al., 1983; Van Nugteren et al., 2009 a). There is

also likely to be a role for priming, whereby the microbial community is activated by addition of a small amount of relatively

bioavailable organic C, allowing remineralisation of more of the ambient, less bioavailable organic C than would otherwise

have occurred (Bianchi, 2011; Van Nugteren et al., 2009 b; Hannides and Aller, 2016). Further experiments that can distinguish

these mechanisms would be informative.

### 4.2 The Fate of Fresh Organic C

In the 13C-AA experiment sufficient $^{13}$C pools were quantified to allow a carbon budget to be calculated (with *Hydrobia*

excluded from macrofaual uptake, as mentioned above). In the control and defaunated treatments a mean of 17.5 ± 5.5 % of

the added $^{13}$C was recovered from biologically processed pools (fauna, bacterial biomass and respiration). The restocked

treatment showed the highest percentage (24.2 %), with particularly high uptake into macrofauna (Fig. 8). The $^{13}$C that could

not be accounted for presumably remained in the sediment, although data are not available to confirm this. It is notable that

the uptake of $^{13}$C into both macrofaunal and bacterial biomass were always higher than $^{13}$C respiration (Fig. 8). This is a

surprising result, which is not obtained in previous isotope tracing experiments from estuarine settings (Woulds et al., 2009;

2016). Previous studies have shown that intertidal and estuarine sites have relatively high biomass as well as active microbial

and macrofaunal communities, but it has not previously been observed that assimilation into macrofaunal and microbial

biomass would exceed respiration to this extent. The observation of assimilation of $^{13}$C exceeding respiration of $^{13}$C suggests

that carbon from fresh detritus may be more likely to be incorporated into biomass, while older ambient organic C tends to be

routed to respiration.



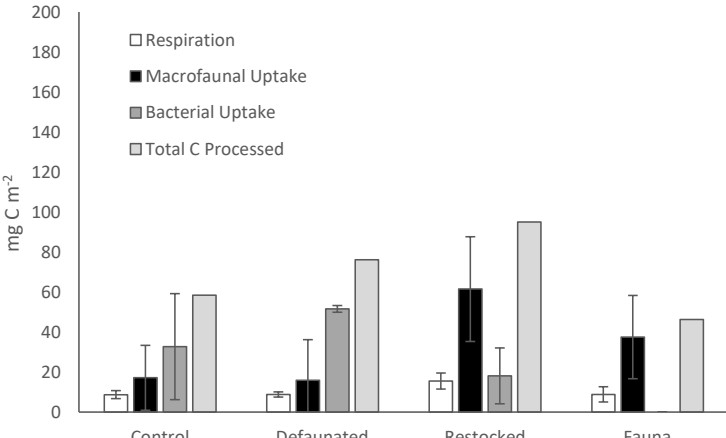

**Figure 8. The distribution of added $^{13}C$ labelled OC between different biologically processed pools in the axenic algae experiment. Note that Hydrobia are not included in macrofaunal uptake due to biomass uncertainties.**

### 4.3 Competition and Functional Redundancy

In both the 13C-AA and 13C-MPB experiments the fauna and defaunated treatments showed similar FOMR rates, measured as production of $^{13}C$-DIC (Figs. 2B, 3). Rates in the restocked treatment were never as high as the sum of the rates when either only fauna or only microbes were present, despite the majority of added C remaining in the sediment.

This suggests that the access to the fresh organic matter may have been the limiting factor on FOMR rates, with the macrofaunal and microbial plus meiofaunal components of the benthic community competing for the resource that they could reach (notably, the same amount of fresh $^{13}C$ was added in each treatment, irrespective of the community biomass present). The fact that a considerable amount of the added C remained in the sediment also indicates that it consisted of different fractions with different bioavailability. In line with competition for the fresh organic C, the uptake of added C into bacterial biomass (Fig. 5) was greatest in the defaunated treatment, while in the control and re-stocked treatments bacterial uptake was supressed by competition with macrofauna. This is consistent with previous studies which have found that the availability of organic matter exerts a control on benthic respiration rates (Provoost et al., 2013), and that, more generally, the functioning of intertidal ecosystems tends to be food limited (Edgar, 1993). Other studies have also suggested that in marine benthic communities, the macrofaunal and bacterial components may compete for detrital organic matter. In two deep sea settings, reduced bacterial



production in the presence of macrofauna has been attributed to competition for organic matter and resource partitioning

(Hunter et al., 2012; 2013). Macrofauna are more able than microbes to locate and exploit concentrated food deposits on the

sediment surface (Van Nugteren et al., 2009 a), and are thought to interact with meiofauna regarding organic matter

availability, although it is not clear whether this includes competition (Schwinghamer et al., 1983), or enhances its availability

to meiofauna through redistribution (Braeckman et al., 2011). Overall, the competition for resources between organisms of

different kingdoms is poorly studied in marine sediments, despite the suggestion that microbes versus eukaryotes may represent

the most prevalent form of competition on Earth (Hochberg and Lawton, 1990).

The differences in FOMR rates between treatments may also be discussed in terms of functional redundancy within the benthic

community, such that fresh organic matter is remineralised at approximately the same rate, irrespective of the identity and (to

some extent) biomass of the organisms present. The 'redundancy' hypothesis for ecosystem functioning (Walker, 1992) states

that an ecosystem function will be delivered by the pool of species in an ecosystem, such that if one species is removed, the

function will be taken over by other species. In the case of our experiments this redundancy could be related to a release from

competition when some organisms are not present. Functional redundancy stands in contrast to the 'rivet' hypothesis (Ehrlich

and Ehrlich, 1991), in which every species in an ecosystem supplies a unique function, such that the removal of any one species

leads to a loss of function. Evidence for redundancy within marine benthic communities has been found previously. For

example, in a cockle removal study (Cesar and Frid, 2009), ecosystem function as measured by sediment surface chlorophyll-

a and organic matter concentrations remained unchanged, despite a shift in the biological traits of the macrofaunal community.

Also, following defaunation of an intertidal site, the carbon flows from microphytobenthos and bacteria into macrofauna

recovered months before the full macrofaunal diversity had re-established (Rossi et al., 2009). On an intertidal mudflat,

manipulations of species richness were found not to impact any ecosystem functions apart from sediment oxygen consumption

(Bolam et al., 2002). This latter effect was thought to be because one species, when present, appeared to have a

disproportionately large role in sediment oxygen consumption, and so could be termed a keystone species. Clarke and Warwick

(1998) analysed macrofaunal communities from two coastal sites and determined that they contained up to 4 sub-sets of

species, each of which alone could deliver the same function as the whole community.

The studies detailed above consider functional redundancy only within macrofaunal communities, and functional redundancy has also been observed within microbial communities (Franklin and Mills, 2006). However, the redundancy suggested by our

experiments is between macrofauna and microbes for fresh organic matter remineralisation. As with competition, redundancy between kingdoms is rarely considered. One macrofauna removal study found that defaunated patches showed reduced ammonium flux and reduced gross primary production (Lohrer et al., 2010), indicating lack of functional redundancy between kingdoms. Other studies which consider the recovery of whole benthic community function after disturbance have found that microbial communities recover very rapidly (over 1-2 days), and are possibly only reliant on the resumption of normal redox

conditions. Thus the opportunity to examine their role in functional redundancy during ecosystem recovery has perhaps been limited (Rossi et al., 2009; Larson and Sundback, 2012).

**4.4 Contrasting Ambient and Fresh Organic C Remineralisation**

In our experiments TOU reflects remineralisation of all organic C present, including ambient sedimentary organic C, while FOMR reflects remineralisation of only the added, fresh organic C. Thus a comparison of TOU and FOMR rates can inform

on the factors controlling remineralisation of different pools of organic C. Ratios of TOU/FOMR (19-1100, Fig. 4) were very high compared to the value of ~1.3 for mineralisation of Redfield Ratio organic matter, and compared to the values of 0.8-2.0 reported by Alongi et al. (2011). This suggests that the majority of the $O_2$ consumption we observed was associated with remineralisation of pre-existing, ambient sedimentary organic C, rather than the $^{13}C$ which was added as fresh algal detritus or MPB. The TOU:FOMR ratios were higher in the restocked treatment than in the other treatments for both experiments (Fig.

4). This suggests that stimulation of ambient C remineralisation occurred by the inverted microbial loop when macrofauna and microbes are both present, but stimulation of mineralisation of fresh organic C did not occur to the same extent. As summarised in a conceptual model in Figure 9, we suggest that there is a marked difference in operation of the inverted microbial loop (Middelburg, 2018) between remineralisation of different fractions of organic matter. We suggest that the inverted microbial loop works to stimulate the degradation of total sediment organic matter, but does not operate on the degradation of newly

deposited, fresh organic matter.





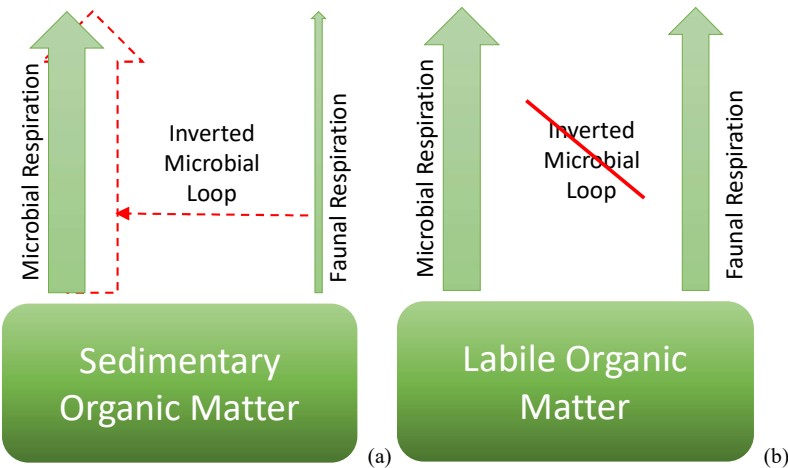

**Figure 9. Conceptual model for the operation of the inverted microbial loop for a) relatively refractory sedimentary organic matter, and b) fresh, labile organic matter, after Middelburg (2018).**

### 4.5 Partitioning Respiration

Our experiments provide a rare empirical quantified partitioning of sediment respiration between different components of the sediment biological community. Fauna-only respiration (measured as TOU and FOMR) was similar to the respiration measured in the defaunated treatment, which represented only the microbial and meiofaunal communities (Figs. 1 and 3). This implies that, independently, these two compartments of the benthic community make approximately equal contributions to total sediment respiration. This contrasts with previous findings that bacteria dominate sediment respiration (Hubas et al., 2006),

production (Schwinghamer et al., 1986), and organic matter degradation (Lillebo et al., 1999). Herman et al. (1999) estimated that macrofauna contributed 15-20% of SCOC. In contrast, on the macrofauna rich Goban Spur, Heip et al. (2001) calculated that macrofauna accounted for a greater proportion of community respiration than bacteria. Thus the results of this study are relatively unusual but not unprecedented, and support a recent suggestion that direct C metabolism by fauna should now be included in diagenetic models (Middelburg, 2018). This is especially the case for shallower (i.e. coastal, shelf, and some

continental margin) sediments where biomass tends to be high (Wei et al., 2010; Stratmann et al., 2019).



## 5 Conclusions

The experiments reported here provide a direct examination of the inverted microbial loop concept for marine sediments as proposed by Middelburg (2018). Our specific findings are that:

- The inverted microbial loop, in which macrofaunal processes stimulate microbial activity, was demonstrated to
influence the remineralisation of total sediment organic matter, as revealed by $O_2$ consumption rates.

- Macrofauna and the microbial community appeared to compete for the added, fresh organic matter, and this was a limiting resource when both communities were present together.

- Partitioning of total respiration between fractions of the benthic community showed that the direct contribution by macrofauna can be of a similar magnitude than that of the microbial community.

**Availability of data and samples** – Data available through University of Leeds Data Repository, DOI TBC. Samples are not available.

**Authors' contributions** All authors contributed to the experimental design. The experiments were performed by CW and SHM. The manuscript was written by CW and DvO, with contributions from other co-authors.

**Competing interests:** The authors declare that they have no conflict of interests.

**Acknowledgements**

The authors would like to thank Pieter Van Rijswijk and Bert Sinke for their assistance in the field and laboratory. The experiments were conducted at the Royal Netherlands Institute for Sea Research (NIOZ) in Yerseke.



**Financial support** This work was funded by a VIDI grant (no. 864.08.004 from NWO, The Netherlands) to Filip Meysman and a Royal Society (UK) travel grant to Clare Woulds.

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
