# Peer review of "Loop Stimulates Mineralisation Microbial Inverted of Sedimentary Organic Detritus"

_EGUsphere, 2025_

## Referee Comment (RC2)

**Review report**

**The Inverted Microbial Loop Sedimentary Organic Detritus**

Clare Woulds, Dick van Oevelen, Silvia Hidalgo-Martinez, Filip J. R. Meysman

**Summary of the manuscript**

The manuscript takes an experimental approach to investigate the degradation of sedimentary detritus, specifically examining the role of microbes in the presence of macrofauna during the degradation process. The authors track respiration by quantifying oxygen consumption and dissolved inorganic carbon in control, defaunated, and faunated sediment samples. The study is framed to test the inverted microbial loop hypothesis proposed by Middelburg (2018), which suggests that macrofauna mix freshly deposited, labile organic matter into deeper sediment layers, where it becomes available for microbial metabolism, this process stimulating microbial respiration and enhancing decomposition, including the breakdown of otherwise refractory organic material. The authors find that macrofaunal and microbial communities contribute approximately equally to total community respiration and that macrofauna and microbes compete for fresh organic matter. Their findings partly confirm the "inverted microbial loop," demonstrating that macrofaunal respiration can be of a similar magnitude to that of the microbial community in benthic respiration.

**Major review**

The manuscript investigates an interesting question on the metabolism of sedimentary organic matter and uses an appropriate experimental approach to test the hypothesis in question, drawing noteworthy conclusions. My review mainly highlights two major weaknesses that, if addressed, would improve the readability, clarity of results, and potentially allow the study to reach a wider audience.

First, the style of the manuscript writing assumes substantial domain expertise on the part of the reader. There is very little introduction of terms or clear progression from the explanation of terminology to the broader context of the study and the relevance of the questions outside of benthic dynamics. There are few connection points. For example, while the study is focused on the "inverted microbial loop," there is no prior introduction of the microbial loop before presenting the inverted microbial loop. In addition, the text contains a lot of jargon which, although obvious to domain experts, is not common knowledge. I point out a few examples below. Making the first two paragraphs of the introduction as plain as possible, and taking the time to introduce the main concepts, would make the manuscript much easier to read.

Second, while the manuscript presents a lot of valid information, it lacks a structure that holds

the text together and draws appropriate summaries when necessary. For example, both the abstract and conclusions lack a clear summary and discussion of the implications of the study. Moreover, parts of the discussion read more like results; for instance, sections 4.4–4.5 seem to report findings rather than discuss their implications in the context of the literature. This kind of writing makes the text quite dense and not easy to digest, thereby narrowing the readership to domain experts only.

Finally, I would like to offer a few suggestions that could help provide additional context, even if not essential. Benthic processes are generally a missing component in Earth system models; the last time I checked, only the Norwegian model (NorESM) and GFDL-Cobalt included benthic processes. As such, most modeled benthic processes remain elementary. An improved understanding of these processes would be very helpful for modeling studies. Moreover, proposed carbon dioxide removal interventions, such as kelp harvesting, are likely to have significant impacts on benthic respiration and oxygen. Studies such as yours provide a valuable framework for designing insightful studies on how benthic ecosystems and biological processes might be affected by these interventions.

**Minor comments**

Line 20: An odd sentence to end an abstract. Either include a concluding sentence, a summary of your results, or the implications of your study.

Line 45: "In contrast to our knowledge of the external factors that govern total community respiration, we lack an understanding of the internal mechanisms that determine how respiration is partitioned amongst the different groups of organisms that make up the benthic community, and especially, how interactions between those groups can influence the total community respiration. The microbial component is often assumed to be of paramount importance in driving total community respiration, and evidence for this comes from both observational (e.g., Schwinghamer et al., 1986; Hubas et al., 2006) and modelling (Van Oevelen et al. 2006) studies." This section provides a strong introduction to the study. I recommend introducing a brief version of it earlier and then expanding on it here.

Line 65: Terms like bioirrigation are not common vocabulary; I would introduce these. Also, consider contrasting the standard microbial loop with the inverted microbial loop. I would also define labile and refractory organic carbon.

Line 90: "more refractory and 'diffusively' distributed sedimentary organic C, and less to the fresh organic C that is concentrated on the sediment surface." I would expand on this; it is unclear what is meant by "diffused" here.

Figure 1: Remind the reader what the 13C-AA experiment is, and what the 13C-MPB experiment is, in the figure caption.

Line 360: I would include a variation of this text in the conclusions, to better place the findings of your study in context.

Sections 4.4 and 4.5 read like results rather than discussion.

Conclusions section: Now that we know your results, so what? What have we learned, and what future relevant questions arise in the context of your findings?